# Real-Time Guitar Amplifier Emulation with Deep Learning †

**Alec Wright [1],\* , Eero-Pekka Damskägg [2], Lauri Juvela [2] and Vesa Välimäki [1]**

1   Acoustics Laboratory, Department of Signal Processing and Acoustics, Aalto University, FI-02150 Espoo, Finland; vesa.valimaki@aalto.fi
2   Neural DSP Technologies, FI-00150 Helsinki, Finland; eero-pekka@neuraldsp.com (E.-P.D.); lauri@neuraldsp.com (L.J.)
\*   Correspondence: alec.wright@aalto.fi
†   This paper is an extended version of our paper published in the Proceedings of the 16th International Sound and Music Computing Conference SMC-19 in Malaga, Spain, 28–31 May 2019.

**Abstract:** This article investigates the use of deep neural networks for black-box modelling of audio distortion circuits, such as guitar amplifiers and distortion pedals. Both a feedforward network, based on the WaveNet model, and a recurrent neural network model are compared. To determine a suitable hyperparameter configuration for the WaveNet, models of three popular audio distortion pedals were created: the Ibanez Tube Screamer, the Boss DS-1, and the Electro-Harmonix Big Muff Pi. It is also shown that three minutes of audio data is sufficient for training the neural network models. Real-time implementations of the neural networks were used to measure their computational load. To further validate the results, models of two valve amplifiers, the Blackstar HT-5 Metal and the Mesa Boogie 5:50 Plus, were created, and subjective tests were conducted. The listening test results show that the models of the first amplifier could be identified as different from the reference, but the sound quality of the best models was judged to be excellent. In the case of the second guitar amplifier, many listeners were unable to hear the difference between the reference signal and the signals produced with the two largest neural network models. This study demonstrates that the neural network models can convincingly emulate highly nonlinear audio distortion circuits, whilst running in real-time, with some models requiring only a relatively small amount of processing power to run on a modern desktop computer.

**Keywords:** acoustic signal processing; audio systems; music; nonlinear systems; signal processing algorithms; supervised learning

## 1. Introduction

Many popular guitar amplifiers and distortion effects are based on analog circuitry. To achieve the desired distortion of the guitar signal, these circuits use nonlinear components, such as vacuum tubes, diodes, or transistors. As music production becomes increasingly digitised, the demand for faithful digital emulations of analog audio effects is increasing [1–3]. The main objective of the field of Virtual Analog (VA) modelling is to create digital emulations of these analog systems. This allows bulky, expensive and fragile analog equipment to be replaced by software plugins that can be used on a modern desktop computer.

A common approach for VA modelling of distortion effects is "white-box" modelling [4–7]. White-box modelling is based on analysis and discrete-time simulation of the analog circuitry. If the circuit and the characteristics of its nonlinear components are known, white-box modelling can be very accurate. However, circuit simulation can be computationally demanding when there are many

reactive components and nonlinear elements in the circuit, and the involved design process can be labor intensive.

An alternative approach for VA modelling is "black-box" modelling. Black-box modelling is based on measuring the circuit's response to some input signals, and creating a model which replicates the observed input-output mapping. Black-box models for VA modelling include block-oriented models, which are based on assumptions about the design of the modelled circuit [8–12]. As an example, a Wiener model [8,11] emulates the circuit as a linear filter followed by a static nonlinearity. Other black-box modelling methods include Volterra series models [13,14] and dynamical convolution [15]. Approaches which fall somewhere in between white and black box are known as "grey-box" models. These use knowledge of the circuit to create the model structure, but also require input/output data measured from the device being modelled.

In recent years, more machine learning based models have been proposed for nonlinear circuit modelling. One approach has been to use a state space representation of the circuit, and train a machine learning algorithm to predict both the circuit's states and output, using, for example, kernel regression [16] or a neural network [17]. These are examples of grey-box models, as analysis was required to determine the number and location of the circuit states.

In the past year, a number of neural network models have been suggested for black-box vacuum tube amplifier emulation. Zhang et al. [18] proposed a long short-term memory (LSTM) model, with many layers but a small hidden size in each layer, although the authors reported clearly audible differences between the resulting model and the target device. Schmitz and Embrechts have proposed a hybrid convolutional and recurrent model [19], as well as number of other recurrent, dense and convolutional networks [20]. In [21] we presented a single layer recurrent model along with a real-time C++ implementation. Additionally, in [22] the authors presented a convolutional neural network which was used for modelling several digital nonlinear audio effects.

This paper focuses on a model we proposed [23] for nonlinear cicuit black-box modelling, that was based on the WaveNet convolutional neural network [24]. The neural network model is made up of a series of convolutional layers, with each layer consisting of a dilated filter followed by a dynamically gated nonlinear activation function. In [23] listening tests were conducted to validate the performance of the model when emulating the Fender Bassman tube amplifier. Following from this, we presented models, with a focus on real-time performance, of three guitar distortion effects: the Ibanez Tube Screamer, the Boss DS-1, and the Electro-Harmonix Big Muff Pi [25].

This paper is an extended version of our previous work [25]. To further validate the neural network models, emulations of two vacuum tube guitar amplifiers are presented: the Blackstar HT-5 Metal, and the Mesa Boogie 5:50 Plus. An existing guitar amplifier dataset [26], was used to train both the WaveNet-style network, as well as a variant of the Recurrent Neural Network (RNN) model mentioned previously [21]. Listening tests were conducted to evaluate the perceptual quality of the neural network emulations, and real-time C++ implementations of both the RNN and WaveNet-style neural networks were used to measure the processing speed of the models. This allows for the networks to be compared both in terms of perceptual quality and computational efficiency.

The rest of this paper is structured as follows. Section 2 details the WaveNet-style deep neural network used for black-box modelling in this work, as well describing experiments which were carried out to select appropriate parameters for the model. Section 3 describes the RNN model. In Section 4, the neural network models of the two amplifiers are presented, along with the results of the listening tests. Section 5 concludes the paper.

## 2. WaveNet-Style Model

The original WaveNet [24] is a convolutional autoregressive model, where the previous output sample is fed back to the model for use in making the next prediction. The neural network architecture described in this section is a feedforward variant of the WaveNet neural network.

The model is shown in Figure 1. The neural network consists of a series of convolutional layers. The raw input waveform is given as input to the first convolutional layer. The convolutional layers apply linear filtering and a nonlinear activation function to the signal. Optionally, the output of the network can be conditioned on user controls.

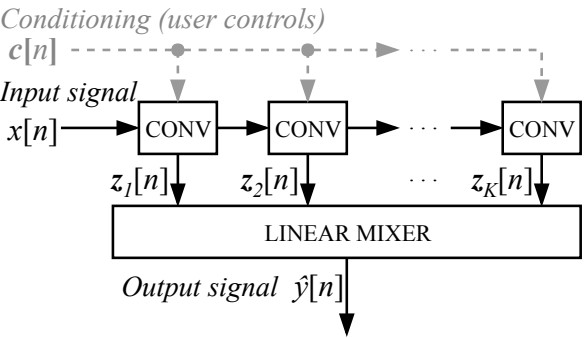

**Figure 1.** Proposed WaveNet-style neural network model.

In previous work [23], the outputs of the convolutional layers were fed to a three layer *post-processing module* with $1 \times 1$ convolutions and nonlinear activation functions. In convolutional neural network terminology, a $1 \times 1$ convolution refers to a matrix multiplication applied at each time step in the signal. In this work, the post-processing module is replaced by a linear mixer, i.e., a single linear $1 \times 1$ convolution layer. Using the linear output layer reduces the complexity and computational load of the network, and, according to our experiments the resulting network performs similarly or better than when using the three layer post-processing module.

### 2.1. Convolutional Layer

The convolutional layer used in the model is shown in Figure 2. The input signal is first processed by the dilated causal FIR filter $H_k(z^{d_k})$, where $k$ is the layer index and $d_k$ is the integer-valued *dilation factor* of the filter. Since the convolutional layers generally have multiple channels, the filtering is performed as a multiple-input and multiple-output (MIMO) convolution with a kernel $H_k$. This means that a filter is learned for each pair of input and output channels. The individual filters in the kernel have impulse responses

$$h[n] = \sum_{m=0}^{M-1} w_m \delta[n - md_k], \tag{1}$$

where $\delta[n]$ is the Kronecker delta function, and $w_m$ are the non-zero coefficients of the filters learned by the network. Next, a nonlinear activation function $f(\cdot)$ is applied to the biased convolution output, producing the layer output

$$z_k[n] = f[(H_k * x_k)[n] + b_k], \tag{2}$$

where $*$ denotes the convolution operator, and $b_k$ is the learned bias term. The layers include a residual connection, which means that the input to the next layer is

$$x_{k+1}[n] = W_k z_k[n] + x_k[n], \tag{3}$$

where the $1 \times 1$ convolution kernel $W_k$ controls the mixing between the layer input $x_k$ and the layer output $z_k$ before the next layer.

Each convolutional layer can be viewed as a Wiener model: a linear filter followed by a static nonlinearity. In previously proposed black-box and grey-box models of musical distortion circuits it has often been assumed that the device can be approximated by a Wiener model [8,11], a Hammerstein model [9] or a Wiener-Hammerstein [10,12] model. As a cascade of Wiener models, the WaveNet-style neural network makes fewer assumptions about the design of the modelled device, and so should be

applicable to the modelling of a broad range of nonlinear systems. Furthermore, the deep learning approach optimizes the system response jointly, and not block-by-block, so not only the model but also the optimization makes fewer assumptions about the behavior of the device under study.

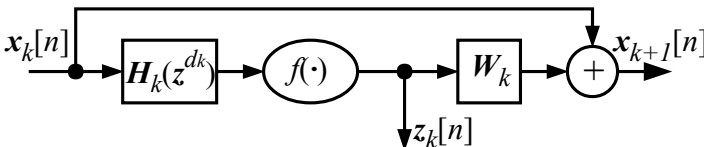

**Figure 2.** Block diagram of a single convolutional layer.

### 2.2. Receptive Field

The WaveNet-style model is a causal feedforward model, as such each output sample predicted by the model, depends only on the $N$ previous input samples, where $N$ is known as the receptive field of the model. The receptive field depends on the number of convolutional layers, and the lengths of the filters in the layers. This is illustrated in Figure 3. The example network has 3 convolutional layers with dilation factors $d_k = \{1, 2, 4\}$, and $M = 2$ non-zero coefficients for each filter. It can be seen that in this case, the current output sample depends on eight latest input samples. That is, the network has a receptive field of $N = 8$. Generally, the receptive field is given by

$$N = (M-1) \sum_{k=1}^{K} d_k + 1,\tag{4}$$

where $K$ is the number of convolutional layers. By increasing the dilation by a factor of two in each layer, the model order can be increased to thousands of samples with relatively few layers, allowing feedforward modelling of systems with long impulse responses.

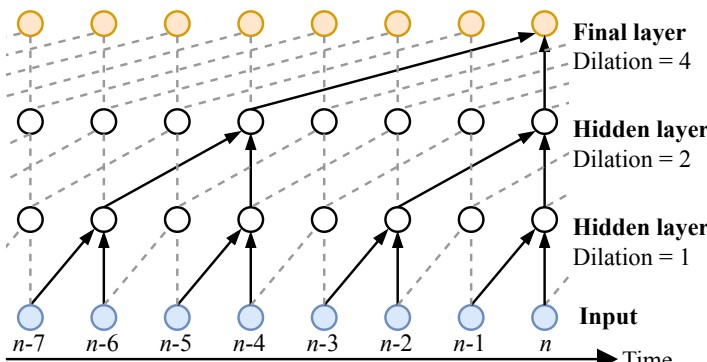

**Figure 3.** Visualization of three convolutional layers in series and the resulting receptive field of $N = 8$. The figure has been adapted from [24].

### 2.3. Real-Time Implementation

The WaveNet-style model was implemented in C++. The real-time application was built using the open source JUCE framework. JUCE allows building cross-platform audio applications as well as VST, AU, and AAX plugins from a single source code. The Eigen library was used for matrix and vector operations. The source code is available at https://github.com/damskaggep/WaveNetVA.

The real-time performance of the C++ code was estimated for several model configurations. The models were tested using an Apple iMac with an 2.8 GHz Intel Core i5 processor, using a sample rate of 44.1 kHz. During the test, all other applications were shut down and the computer was disconnected from the internet. This was done to minimize the effect of other processes in the test.

Figure 4 shows the processing speed of the model with different numbers of layers and different numbers of convolution channels. In this case, the models use a gated activation function.

The processing speed is expressed as a factor of the requirement for real-time application. Clearly, the computational load increases as the number of layers and channels is increased. The largest model running in real time uses 18 layers and 16 channels in the convolutional layers. With 24 layers, a model with 8 convolutional channels can be run in real time.

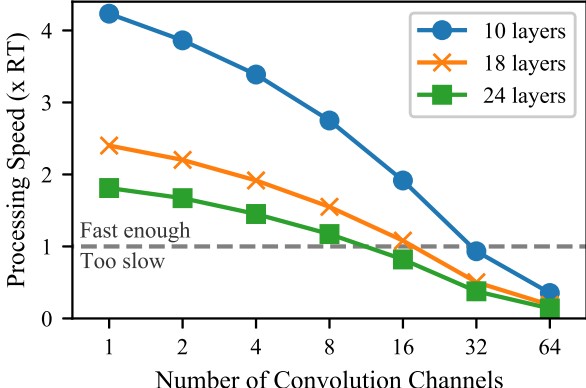

**Figure 4.** The processing speeds of models with different numbers of layers and convolution channels. The models use the gated activation. The cases above the horizontal dashed line can run in real time.

Figure 5 shows the processing speed of the 18-layer model using different activation functions. The activation functions are detailed in Section 2.6. The rectified linear unit (ReLU) is the computationally cheapest and the gated activation is the most computationally expensive activation.

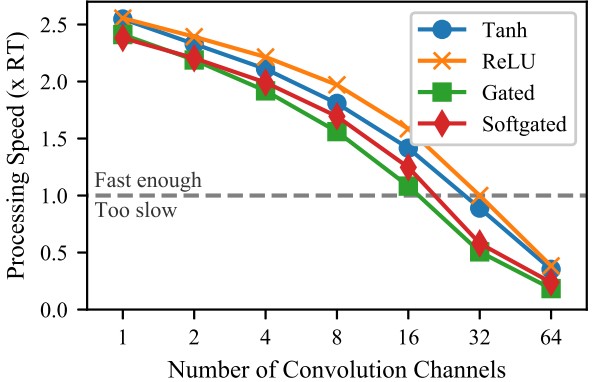

**Figure 5.** The processing speeds of the 18-layer model with different activation functions and different numbers of convolution channels. The cases above the horizontal dashed line can run in real time.

### 2.4. Experiments

To determine suitable parameters for the WaveNet-style model, a number of experiments were conducted, in which models of three guitar distortion effects were created: the Ibanez Tube Screamer, the Boss DS-1, and the Electro-Harmonix Big Muff Pi. Detailed circuit analysis of all three pedals can be found online. Digital models for the Tube Screamer [4,7,11,27], DS-1 [5,27], and Big Muff Pi [11,28] have been proposed previously. More analysis on these pedals can also be found in our previous work [25].

To estimate the required receptive field for modelling of the distortion effects, their linear impulse responses were estimated using the swept-sine technique [9,29]. A low-level sine sweep was used in order to minimize the effect of circuit nonlinearities in the measurement. The estimated lengths of the impulse responses were approximately 35 ms for the Big Muff, and approximately 45 ms for the Tube Screamer and the DS-1. With a 44.1-kHz sample rate, these correspond to required receptive fields of approximately 1500 to 2000 samples, respectively.

### 2.5. Dataset Generation

Training data was generated by processing audio through the three distortion effects pedals. The devices were measured using an audio interface connected to a computer via USB. One output of the audio interface was connected to the input of the measured device. The output of the device was recorded by connecting it to one of the inputs of the audio interface. The output of the audio interface was also directly connected to another input of the audio interface, in order to estimate the effect of the audio interface in the measurement, as suggested in [12]. The recorded direct signal from the audio interface and the recorded output from the device under study make up the input/target pairs used in the training of the network.

The input sounds processed through the device were obtained from the guitar and bass guitar datasets (https://www.idmt.fraunhofer.de/en/business_units/m2d/smt/guitar.html, https://www.idmt.fraunhofer.de/en/business_units/m2d/smt/bass_lines.html) described in [30,31], respectively. A random subset with 5 min of audio was picked from the datasets, with 2.5 min of guitar and 2.5 min of bass sounds. The data generated using these sounds was used for training. An additional minute of audio was randomly selected for validation.

All three modelled devices have a knob to control the intensity of the distortion effect and a "Tone" knob to control the filter in the tone stage. For the measurements, all knobs were set to the 12 o'clock, or middle, position. Filtering occurs in the tone stages of all pedals even when the Tone knob is set to the middle position [32]. That is, the middle position of the knob does not indicate an allpass setting for the filters in the tone stages.

### 2.6. Model Training and Selection

The performance of the neural network depends mostly on the number of channels used in the convolutional layers, the activation function, and the dilation pattern. The choice of these hyperparameters also affects the computational load of the model, as shown in Section 2.3. A hyperparameter search was conducted to find a suitable trade-off between model performance and computational load. Three different dilation patterns were considered:

$$d_k = \{1, 2, 4, \ldots, 512\},$$
$$d_k = \{1, 2, 4, \ldots, 256, 1, \ldots, 256\}, \text{ and}$$
$$d_k = \{1, 2, 4, \ldots, 128, 1, \ldots, 128, 1, \ldots, 128\}.$$

These dilation patterns correspond to models with 10, 18, and 24 convolutional layers, respectively. The number of non-zero coefficients in the filters was set to $M = 3$, which means that, according to Equation (4), the 10, 18, and 24-layer networks have the receptive fields of $N = 2047, 2045$, and 1530 samples, respectively. At the 44.1-kHz sample rate, these receptive fields correspond to approximately 46, 46, and 35 ms, respectively.

For the convolutional layers, the performance of the following activation functions were compared: the hyperbolic tangent:

$$z = \tanh(H * x), \tag{5}$$

the rectified linear unit (ReLU):

$$z = \max(0, H * x), \tag{6}$$

and the gated activation, which was used in the original WaveNet [24]:

$$z = \tanh(H_f * x) \odot \sigma(H_g * x), \tag{7}$$

where $\odot$ is the element-wise multiplication operation, $\sigma(\cdot)$ is the logistic sigmoid function, $H_f$ and $H_g$ are the filter and gate convolution kernels, respectively. Finally, the softsign-gated activation, as used in [33], was evaluated:

$$z = g(H_f * x) \odot g(H_g * x), \tag{8}$$

where the hyperbolic tangent and the logistic sigmoid of the standard gated activation are both replaced by the softsign function:

$$g(x) = \frac{x}{1 + |x|}. \tag{9}$$

The softsign nonlinearity can be computationally cheaper than the hyperbolic tangent and the logistic sigmoid functions, as shown in Section 2.3, while having a similar shape.

With the gated activations, the convolutional layer used in the model can no longer be considered a Wiener model. Instead, it can be described as two parallel Wiener models, whose outputs are multiplied together to produce the layer output.

## 2.7. Training and Loss Function

The neural network models in this paper were trained by minimizing the error-to-signal ratio (ESR) with respect to the training data. During training and validation, a pre-emphasis filter was applied to the output and target signals before computing the ESR. For a signal of length $N$, the pre-emphasised ESR loss is given by:

$$\mathcal{E}_{\mathrm{ESR}} = \frac{\sum_{n=0}^{N-1} |y_p[n] - \hat{y}_p[n]|^2}{\sum_{n=0}^{N-1} |y_p[n]|^2}, \tag{10}$$

where $y_p$ is the pre-emphasised target signal and $\hat{y}_p$ is the pre-emphasised output of the neural network. The denominator in the ESR normalises the loss with regards to the energy of the target signal, preventing the loss from being dominated by the segments of signal with higher energy.

For the WaveNet hyperparameter selection, the loss function was used with the following first-order high-pass pre-emphasis filter:

$$H(z) = 1 - 0.95z^{-1}. \tag{11}$$

Pre-emphasis filtering is used to emphasise certain frequencies in the loss function. The high-pass pre-emphasis filter was intended to boost the mid and high frequencies, as our preliminary experiments indicated that without the pre-emphasis filtering the resulting neural network produced audible errors in the mid to high frequency range. The models were trained using the Adam optimiser [34]. The validation error was computed after each epoch. Early stopping was used with a patience of 20 epochs. The training data was split into 100 ms training examples, and a mini-batch size of 40 was used. A sample rate of 44.1 kHz was used in the experiments.

## 2.8. Results

In the following, the results of the hyperparameter search are presented. As there is an interest in the real-time performance of the models, the validation loss is shown as a function of the processing speed on the developed real-time C++ implementation of the model.

The effect of the choice of dilation pattern on the validation loss is shown in Figure 6. The validation loss is reported as an average loss over all the modelled devices. All models shown in Figure 6 use the gated activation, as given by Equation (7). The number of convolution channels was varied with values 2, 4, 8, 16, and 32.

It can be seen that the 10-layer model performs favorably with respect to the processing speed, while still obtaining a relatively low ESR. The 10-layer model with 16 channels has an average ESR of 4.2%, and runs 1.9 times faster than real time. The 18-layer model with 16 convolution channels has the lowest ESR of the models which run faster than real time. The model has an average ESR of 3.1%, and runs 1.1 times faster than real time.

Overall, the 24-layer model performs more poorly than the 18-layer model. It is possible that this is because the receptive field of the 24-layer model is slightly shorter than the estimated impulse response lengths of the Ibanez Tube Screamer and the Boss DS-1.

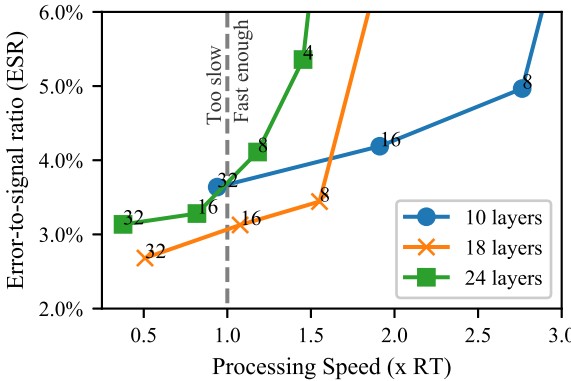

**Figure 6.** The validation error-to-signal ratio (ESR) as a function of the processing speed, using different numbers of layers and convolution channels. All models shown use the gated activation. The number of convolution channels used is indicated next to each model.

The effect of the choice of activation function is shown in Figure 7. The models shown in Figure 7 use 18 layers and the number of convolution channels was again varied with values 2, 4, 8, 16, and 32. It can be seen that the hyperbolic tangent activation performs worst out of all activations. The other activation functions perform similarly to each other. This suggests that the ReLU, the softsign-based gated activation and the standard gated activation are all viable options for real-time applications.

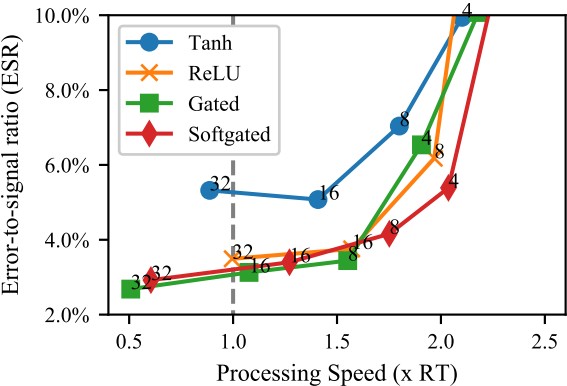

**Figure 7.** The validation error-to-signal ratio (ESR) as a function of processing speed with different activation functions and different numbers of channels in the convolutional layers. The number of convolution channels used is indicated next to each model.

Based on the hyperparameter search, three models were chosen for the final validation, which is presented in Section 4. The hyperparameters of the selected models are shown in Table 1. Only models which run faster than real time were selected. WaveNet1 is the fastest of the selected models, and it has the worst ESR on the validation data. WaveNet3 is the slowest model, and it has the best ESR on the validation data. WaveNet2 is an intermediate model.

**Table 1.** Hyperparameters of selected neural networks.

| Model | WaveNet1 | WaveNet2 | WaveNet3 |
|---|---|---|---|
| Activation | Gated | Gated | Gated |
| Layers | 10 | 18 | 18 |
| Channels | 16 | 8 | 16 |

*2.9. Training Data Length*

An interesting question regarding neural networks for virtual analog modelling is the amount of data required for training a model. To assess the effect of the amount of training data, models were trained with different amounts of training data, and the effect on the validation loss was examined. The results are shown in Figure 8 for WaveNet3, the largest of the selected models. The results are averaged across the three modelled devices.

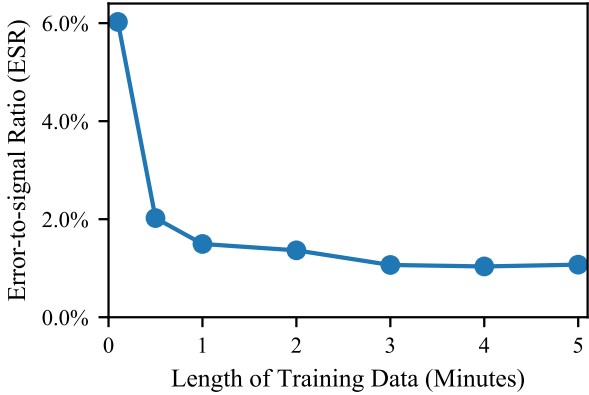

**Figure 8.** The validation energy-to-signal ratio (ESR) with different amounts of training data.

The validation loss decreases as the amount of training data is increased from 10 s to 3 min. Increasing the amount of training data past 3 min appears to have no significant effect on the validation loss.

## 3. Recurrent Neural Network Model

In addition to the WaveNet-style model described in the previous section, an RNN model was also tested during this work. We used this model in our previous work [21], where it was shown to perform similarly to the WaveNet-style model in terms of accuracy, whilst offering a considerable improvement in terms of processing speed.

The model is shown in Figure 9. The neural network is comprised of a single LSTM unit, followed by a fully connected layer. At each time step, a single sample of the raw input waveform is input to the LSTM unit. The output of the LSTM unit is fed into a fully connected layer, to produce a single output sample, which represents the RNN's predicted output for that time step. For this work, the model proposed in [21] is modified slightly, to include a residual connection. The residual connection simply adds the input sample value to the output of the RNN model at each time step. In this configuration the network just needs to learn to predict the difference between the input and output samples.

Like the WaveNet-style model, the RNN model can also be conditioned on user controls. In our earlier work [21], the "Tone" and "ISF" control settings were included in the RNN models of the Big Muff Pi pedal and the HT-1 valve amplifier respectively. For this work conditioning was not included in the RNN models.

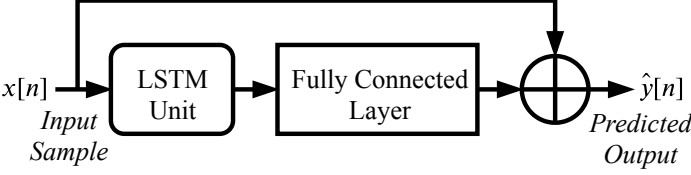

**Figure 9.** Recurrent neural network model.

### 3.1. Long Short-Term Memory

A key difference between a feedforward network, such as the WaveNet-style model, and an RNN model, is that the RNN model has a state. The state is used and updated at each time step. This means that the RNN can operate with just a single sample as input at each time step, whilst still using information from previous time steps.

In this RNN model, the state is part of an LSTM unit. The LSTM was first proposed in [35], its unit state is made up of two vectors, the *hidden state*, $h$, and the *cell state*, $c$. At each time step the input sample, $x[n]$, is used along with the initial unit states, $h[n-1]$ and $c[n-1]$, to produce the LSTM outputs, which are the updated unit states, $h[n]$ and $c[n]$. The hidden state, $h[n]$, is the LSTM's output which in our model is input to the fully connected layer, to produce the RNN's predicted output for that time step. The following functions describe how the input and initial states are used to update the LSTM unit's state.

$$i[n] = \sigma(W_{ii}x[n] + b_{ii} + W_{hi}h[n-1] + b_{hi}), \tag{12}$$

$$f[n] = \sigma(W_{if}x[n] + b_{if} + W_{hf}h[n-1] + b_{hf}), \tag{13}$$

$$\tilde{c}[n] = \tanh(W_{ic}x[n] + b_{ic} + W_{hc}h[n-1] + b_{hc}), \tag{14}$$

$$o[n] = \sigma(W_{io}x[n] + b_{io} + W_{ho}h[n-1] + b_{ho}), \tag{15}$$

$$c[n] = f[n]c[n-1] + i[n]\tilde{c}[n], \tag{16}$$

$$h[n] = o[n]\tanh(c[n]), \tag{17}$$

where $i[n]$ is the input gate, $f[n]$ is the forget gate, $\tilde{c}[n]$ is the candidate cell state, $o[n]$ is the output gate, $\tanh(.)$ is the hyperbolic tangent function and $\sigma(.)$ is the logistic sigmoid function. The PyTorch machine learning library was used to implement the RNN model described in this paper, and (12)–(17) have been reproduced from the PyTorch documentation [36]. The weight matrices and bias vectors in these equations contain the LSTM learnable parameters, which are learned during training.

The size of both the hidden and cell states is equal to the LSTM's *hidden size*, which is a hyperparameter chosen by the user. Increasing the hidden size generally results in the model being more accurate, however it increases the number of learnable parameters in the network, as well as the processing power required to run it.

### 3.2. Fully Connected Layer

The hidden state of the LSTM unit, $h[n]$, is used as input to the fully connected layer. As the fully connected layer is not followed by an activation function, the output of the fully connected layer is simply an affine transformation of the LSTM hidden state. This output is summed with the input sample, $x[n]$, to produce the RNN's predicted output:

$$\hat{y}[n] = W_{fc}h[n] + b_{fc} + x[n], \tag{18}$$

where $W_{fc}$ and $b_{fc}$ are the fully connected layer's weight matrix and bias vector respectively and $h[n]$ is the recurrent unit hidden state at time $n$. The fully connected layer outputs a single value at each time step, and as such it consists of a single neuron.

### 3.3. Real-Time Implementation

In the paper where the RNN model was originally proposed [21], a C++ implementation of the RNN was presented. The real-time application was built using the open source JUCE framework. The Eigen library was used for matrix and vector operations. The processing speed of the RNN model for various hidden sizes is shown in Figure 10.

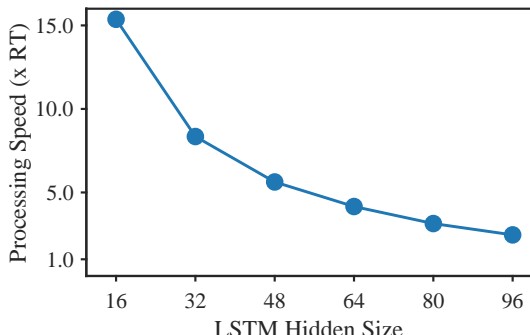

**Figure 10.** The processing speeds of RNN models plotted against the LSTM hidden size.

## 4. Validation

To further validate and compare the WaveNet-style model described in Section 2 and the RNN model described in Section 3, models were created of two valve guitar amplifiers: the Blackstar HT-5 Metal (HT5M), and the Mesa Boogie 5:50 Express Plus (Mesa 5:50). To create these models we used data from an existing dataset [26], which is described in the next section. For both amplifiers, three WaveNet-style models were created using the three hyperparameters indicated in Section 2.8 and shown in Table 1, and three RNN models were created, with LSTM hidden sizes of 32, 64 and 96 respectively.

### 4.1. Training Data

Data from a guitar amplifier dataset [26] was used to train the amplifier models. The dataset input audio consists of approximately three minutes of guitar audio, recorded at a sampling rate 44.1 kHz. For each amplifier in the dataset, the audio is processed ten times, with the amplifier "gain" setting being varied from 1 to 10. The models were trained to emulate the "Blackstar HT Metal 5" on the "Overdrive" channel with the "gain" control set to 10, and the "Mesa Boogie 5:50 Plus" on the "Crunch" channel with the "gain" control set to 3.

The dataset was divided into a training, validation and test set. The "chromatic scale", "chords" and "bamba song" samples made up the training set, whilst the "blues song" and "blues scale Am" samples made up the validation and test sets respectively. More details of the audio in the dataset can be found in [26]. This training set consists of 2 min and 43 s of audio, which is significantly shorter than that which was used for modelling of the distortion effects in Section 2, however, earlier tests, discussed in Section 2.9 and shown in Figure 8, indicate that the length is still sufficient for our purposes.

### 4.2. Loss Function and Training

The tests in Section 2 and in previous work [21,23] used a first-order high-pass pre-emphasis filter. For the experiments presented in this section we use a perceptually motivated pre-emphasis filter based on a low-passed A-Weighting filter, which was first proposed in and tested in [37]. The frequency response of this pre-emphasis filter is shown in Figure 11. The purpose of this filter is to emphasise the frequencies in the loss function, based on their perceived loudness.

As proposed in [21] and used in [37], an additional loss function representing the difference in DC offset between the target and neural network output was also included:

$$\mathcal{E}_{\text{DC}} = \frac{|\frac{1}{N}\sum_{n=0}^{N-1}(y[n] - \hat{y}[n])|^2}{\frac{1}{N}\sum_{n=0}^{N-1}|y[n]|^2}. \tag{19}$$

The final loss function used for training is given by the sum of these two loss functions:

$$\mathcal{E} = \mathcal{E}_{\text{ESR}} + \mathcal{E}_{\text{DC}}. \tag{20}$$

All models were trained using the Adam optimiser [34]. The WaveNet and the RNN models were trained for 1500 and 750 epochs respectively. The validation loss was calculated every other epoch. Once training was complete, the test loss was calculated using the model parameters from the epoch in which the lowest validation loss was achieved.

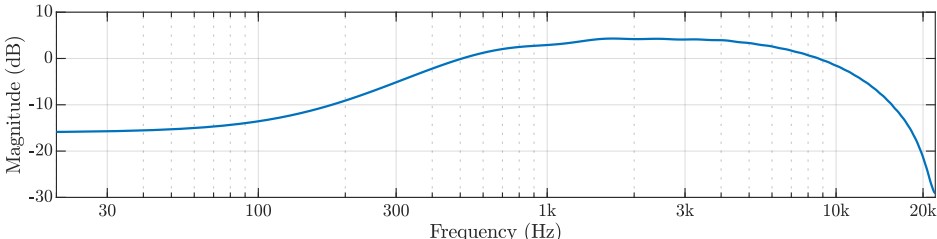

**Figure 11.** The frequency response of the low-passed A-weighting pre-emphasis filter.

### 4.3. Objective Results

Table 2 shows the model processing speeds as well as the ESR of each of the amplifier models, calculated on the test set, without pre-emphasis filtering. For the Mesa 5:50 the loss achieved is low for all the models presented. This indicates that modelling of the less distorting device is easier for the neural network models. For the HT5M the loss is generally significantly higher, however for the two larger WaveNet models a loss of under 1% is achieved. For all models, a lower test loss is achieved on the Mesa 5:50, when compared to the HT5M. This is most likely because the HT5M distorts the signal significantly more than the Mesa 5:50. The WaveNet3 achieved the lowest loss on the HT5M amplifier and performed significantly better than the best RNN model, in terms of loss. The RNN with LSTM hidden size of 96 achieved the lowest loss on the Mesa 5:50, although the differences between the losses achieved by the Mesa 5:50 models are relatively minor.

In terms of processing speeds the RNN models offer a significant advantage over the WaveNet models. Whilst the WaveNet seems to be more accurate for the highly nonlinear HT5M amplifier, it is not clear that the additional cost in processing power justifies the improvement in accuracy.

**Table 2.** Error-to-signal ratio for the test set and model processing speeds.

|  | WaveNet1 | WaveNet2 | WaveNet3 | RNN32 | RNN64 | RNN96 |
|---|---|---|---|---|---|---|
| HT5M | 2.8% | 0.60% | **0.32%** | 4.2% | 3.2% | 1.8% |
| Mesa 5:50 | 0.64% | 0.46% | 0.29% | 0.68% | 0.29% | **0.20%** |
| Speed (x RT) | 1.9 | 1.6 | 1.1 | **8.3** | 4.2 | 2.5 |

### 4.4. Listening Tests

To investigate the impact of the ESR on the perceptual quality of the models, listening tests were conducted. The listening tests were conducted using the multiple stimuli with hidden reference and anchor (MUSHRA) methodology [38]. To avoid presenting test subjects with an excessive number of test conditions, some models were excluded from the tests. The tests included the hidden reference, the WaveNet1 and WaveNet3 models, the RNN models with LSTM hidden sizes of 32 and 96, a multilayer perceptron (MLP) model with an input size of 1024 and 8 hidden layers of 16 neurons each, and an anchor that was created using a static nonlinear function.

The listening test items consisted of six clips, each approximately 2–4 s long, taken from the validation and test sets of the dataset. The MUSHRA test was carried out using the WebMUSHRA interface [39], shown in Figure 12. For each test the user was presented with the reference and the seven test items. For each test the user was asked to rate each of the test items based on "how accurately they emulate the timbre of the reference". The ratings were given on a scale of 0-100. Each of the six test items were processed by the two amplifiers being modelled, resulting in a total of 12 MUSHRA

trials. Audio examples from the MUSHRA test are available online at http://research.spa.aalto.fi/publications/papers/applsci-deep/.

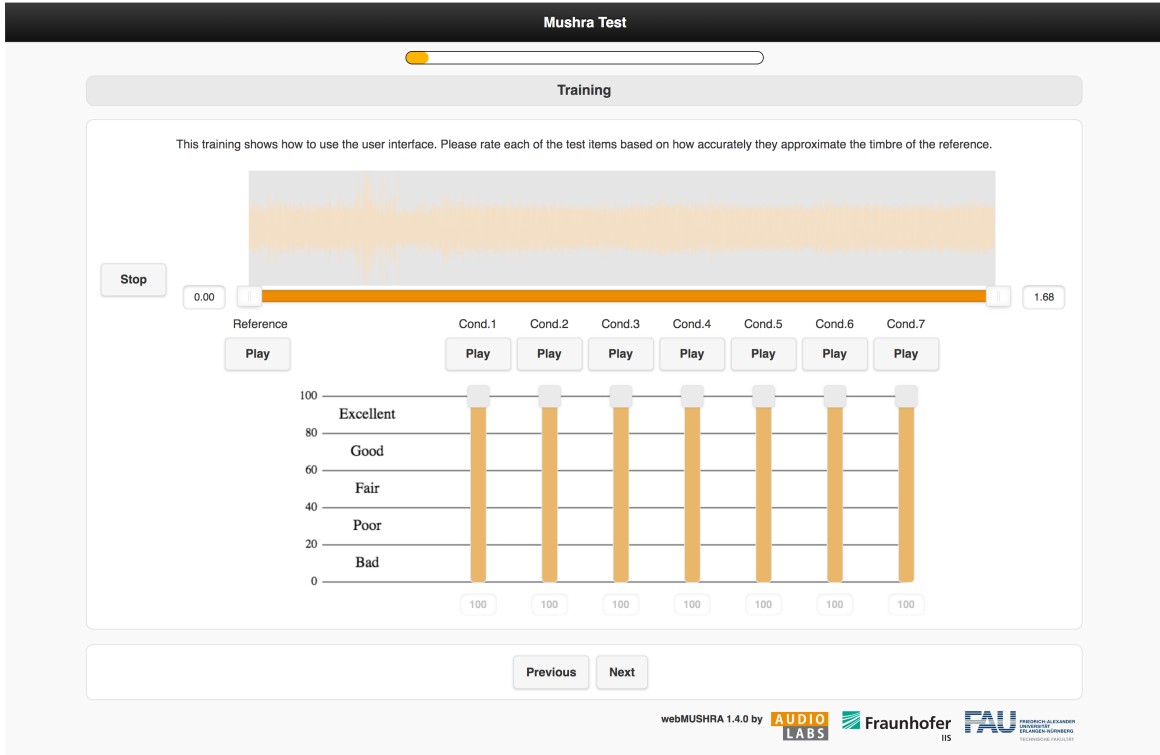

**Figure 12.** The WebMUSHRA interface used during the listening tests.

A total of 14 people completed the MUSHRA test. The subjects all played at least one musical instrument, and none reported any hearing impairments. The listening tests were conducted at the Aalto Acoustics Lab in a silent room using Sennheiser HD-650 headphones. The test subjects were asked to choose a loud, but not uncomfortable, volume during the two training examples at the beginning of the test, and then instructed not to change the volume throughout the rest of the trials. The data of five of the test participants was excluded from the test results as they rated the hidden reference below 90 in more than 15% of the trials.

The results of the MUSHRA trials with 95% confidence interval are shown in Figures 13 and 14 for the Mesa 5:50 and HT5M, respectively. The Mean MUSHRA scores are shown in Table 3. For the Mesa 5:50 the mean MUSHRA scores are all above 90, indicating that the participants generally rated the models as Excellent or better. It is also worth noting that both the WaveNet3 and the RNN-96 received a score of 100 in approximately 80% of the trials conducted, meaning that the subjects were often unable to hear any difference between the models and the reference.

For the HT5M the scores are generally lower, however for the larger WaveNet-style and RNN models the mean score is at or above 90, indicating that they were rated as Excellent on average. The best performing model is the WaveNet3, although as the confidence intervals of the RNN-96 and WaveNet3 perceptual scores are overlapping, it is not clear that the difference between the two is statistically significant or not.

**Table 3.** Mean MUSHRA scores.

|  | **WaveNet1** | **WaveNet3** | **RNN-32** | **RNN-96** |
|---|---|---|---|---|
| HT5M | 75 | **93** | 80 | 90 |
| Mesa 5:50 | 97 | 97 | 93 | **98** |

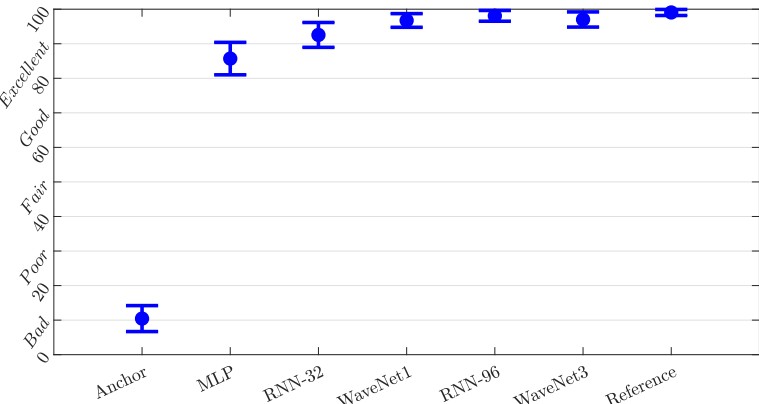

**Figure 13.** Mean results of MUSHRA test with 95% confidence interval, for models of the Mesa 5:50.

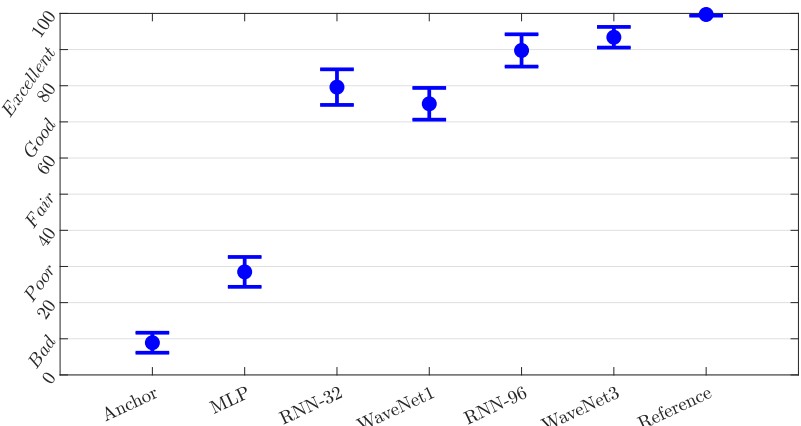

**Figure 14.** Mean results of MUSHRA test with 95% confidence interval, for models of the HT5M.

*4.5. Visual Analysis*

Figure 15 shows a short segment of waveform output by the HT5M amplifier, plotted with the output predicted by the WaveNet3 and RNN-96 models. Additionally, Figure 16 plots the target spectrum produced by the HT5M amplifier, and the spectrum of the WaveNet3 and RNN-96 predictions. The plots in Figure 15a and in Figure 16(middle) clearly show that in this case the WaveNet3 produces a very close match to the target. For the RNN the prediction is not as accurate, as is apparent in Figure 15b and Figure 16(bottom).

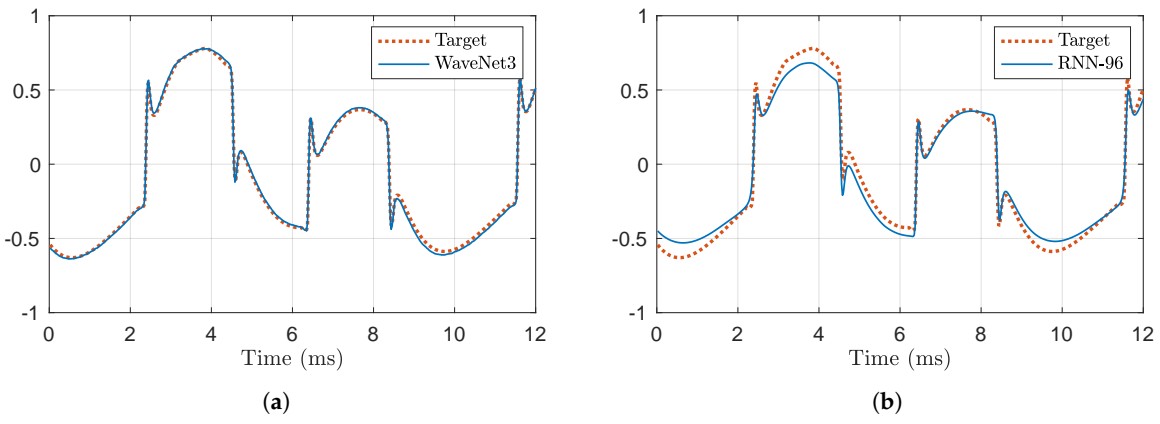

**Figure 15.** Waveforms of amplifier and neural network model outputs. (**a**) Waveform of a guitar sound processed through the HT5M amplifier, and the WaveNet3.; (**b**) Waveform of a guitar sound processed through the HT5M amplifier, and the RNN-96.

In order to estimate the aliasing introduced by the models, Figure 17 shows the spectrum of a 1245-Hz sinusoid fed through the WaveNet3 and RNN-96 models of the HT5M amplifier. It appears that even though the models were trained with non-aliased data, they suffer from aliasing, as was also observed in [25]. At low and middle frequencies, the aliased components are 40 dB softer than the first harmonic component, but at frequencies above about 15 kHz, some aliased components are louder than their nearest non-aliased components. However, while the aliasing is evident with a high-frequency sinusoidal input, no clear aliasing could be heard in the guitar and bass sounds processed through the models. If this becomes a problem, it is possible that aliasing suppression techniques, such as oversampling [40], could be applied during the neural network training, however, this is left for future work.

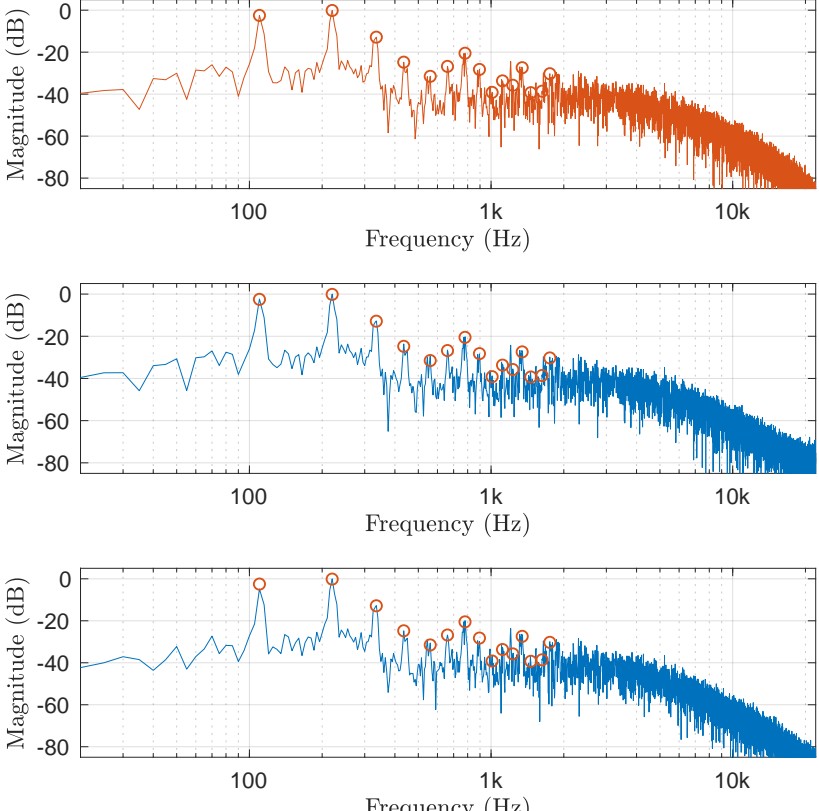

**Figure 16.** Spectrum of a guitar sound processed (**top**) through the HT5M guitar amplifier, (**middle**) through the WaveNet3 model, and (**bottom**) through the RNN-96 model. The circles indicate the level of the first 15 harmonics in the target spectrum.

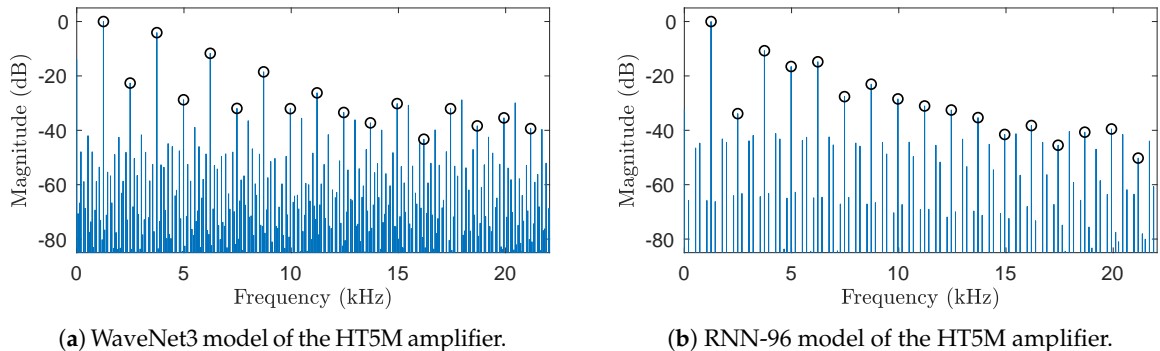

(**a**) WaveNet3 model of the HT5M amplifier.          (**b**) RNN-96 model of the HT5M amplifier.

**Figure 17.** Spectra of a 1245-Hz sinusoid fed through the WaveNet3 and RNN-96 models, the black circles indicate the non-aliased components.

## 5. Conclusions

This work considered the use of deep neural networks for modelling of audio distortion effects. A feedforward variant of the WaveNet neural network was tested as a model for three well-known guitar distortion pedals. Different model configurations were examined to find a suitable compromise between modelling accuracy and computational load. Real-time and low-latency implementations of the deep neural networks were developed.

Models of two popular vacuum tube amplifiers were then created, using the WaveNet-style model, as well as an RNN based model. MUSHRA listening tests were conducted to further validate the neural network models. The results show that both of the neural network types tested can be used to create very perceptually convincing emulations of vacuum tube amplifiers. For the less distorting Mesa 5:50 amplifier, all of the tested models were rated to be Excellent, indicating that even the smaller neural networks tested were sufficiently accurate in this case. For this amplifier, the WaveNet models and the larger RNN were often indistinguishable from the reference. For the more distorting Blackstar HT5M amplifier, only the larger neural network models were generally rated to be Excellent, with the smaller models being judged as somewhere between Good and Excellent on average. In addition to this, all the models included in the listening tests were shown to be capable of running in real-time on a consumer-grade desktop computer, with some of the models only requiring a relatively small amount of the computer's processing power. Possible future work includes investigation and suppression of aliasing that is introduced by the models.

**Author Contributions:** Conceptualization, A.W., E.-P.D., L.J. and V.V.; methodology, A.W., E.-P.D. and L.J.; software, A.W. and E.-P.D.; validation, A.W.; formal analysis, E.-P.D. and L.J.; investigation, A.W. and E.-P.D.; resources, A.W.; data curation, A.W. and E.-P.D.; writing—original draft preparation, A.W. and E.-P.D.; writing—review and editing, L.J. and V.V.; visualization, A.W., E.-P.D. and V.V.; supervision, V.V.; project administration, V.V.; funding acquisition, V.V. All authors have read and agreed to the published version of the manuscript.

**Funding:** This research is part of the "Nordic Sound and Music Computing Network—NordicSMC", NordForsk project number 86892.

**Acknowledgments:** We acknowledge the computational resources provided by the Aalto Science-IT project.

**Conflicts of Interest:** The authors declare no conflict of interest.

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
