# Peer review of "Real-Time Guitar Amplifier Emulation with Deep Learning"

_applsci, doi:10.3390/app10030766_

Round 1

Reviewer 1 Report

Very interesting and well-written article. General approach and specific methodology clearly described, enabling non-expert to understand the work.

The footnote “links” l. 133 are not separated by comma, so it looks like 12.

Could it make sense to briefly explain the interest of adding the pre-emphasis filter?

About equations 12-17, did you come up with these equations by yourselves or do they come from some reference work (to be cited)?

Not clear to me “in the error” (l. 269)

“network” (l. 272) maybe vague? Neural network?

There is redundancy between l.294-296 and l.299-300

Figure 16 bottom is currently an exact copy of middle one.

There are some typos in the conclusion section. “as well an”, “the both”

Don’t forget to remove the acknowledgment section if you don’t use it.

Author Response

Thank you very much for your comments. I have addressed them all now, and a point by point response for each comment is included below. Additionally I have added a few lines to the conclusion (l. 348-353), describing briefly the listening test results.

Comment: 'The footnote “links” l. 133 are not separated by comma, so it looks like 12.'

Response: I have added a comma.

Comment: 'Could it make sense to briefly explain the interest of adding the pre-emphasis filter?'

Response: I have added some justification for the use of pre-emphasis filtering (l. l. 166-169).

Comment: 'About equations 12-17, did you come up with these equations by yourselves or do they come from some reference work (to be cited)?'

Response: I have added a reference to the original LSTM paper (l. 225) and explained that the PyTorch machine learning library was used to implement the LSTM, and that the equations are reproduced from the PyTorch documentation (l. 232 - 234).

Comment: 'Not clear to me “in the error” (l. 269)'

Response: I have changed it to say 'in the loss function'.

Comment: '“network” (l. 272) maybe vague? Neural network?'

Response: I have changed it to 'neural network'.

Comment: 'There is redundancy between l.294-296 and l.299-300'

Response: I have removed the redundant information.

Comment: 'Figure 16 bottom is currently an exact copy of middle one.'

Response: I have fixed Figure 16, so it now shows the correct spectra.

Comment: 'There are some typos in the conclusion section. “as well an”, “the both”'

Response: I have fixed these typos.

Comment: 'Don’t forget to remove the acknowledgment section if you don’t use it.'

Response: I have added an acknowledgement.

Reviewer 2 Report

Title: Real-Time Guitar Amplifier Emulation with Deep Learning
Authors: Alec Wright, Eero-Pekka Damskägg, Lauri Juvela, and Vesa Välimäki

What the paper aims:

This paper investigates the application of deep neural networks in the form of WaveNets and recurrent neural networks for black-box modelling of audio distortion circuits (i.e., guitar amplifiers and distortion pedals). The article claims that three minutes of audio data will be enough for a proper training to emulate nonlinear audio distortion circuits.

Although the authors have previous contributions/work in this area, this paper and the methods are an extension to their previous work, containing novel parts compared to their past studies.

The article has been nicely written and most of the required information are provided. The paper is well structured, and background is well-written Use of figures is very good and add to understanding of the data and methods. State of the art was well explained and helped to put the paper into context. Numbers of samples used for training/validation of the nets are satisfactory. The methods are appropriate and well explained. The conclusions are consistent with the evidence and arguments The results support the hypothesis and conclusions.

Author Response

Thank you for your review and comments. I have made some minor changes to the paper in addressing some grammatical errors, as well as updating one of the figures and adding some lines to the conclusion (l. 348-353).